# JMJD6 cleaves MePCE to release positive transcription elongation factor b (P-TEFb) in higher eukaryotes

Schuyler Lee[1,2†], Haolin Liu[1,2†], Ryan Hill[3], Chunjing Chen[4], Xia Hong[1,2], Fran Crawford[1], Molly Kingsley[5,6], Qianqian Zhang[7], Xinjian Liu[8], Zhongzhou Chen[7], Andreas Lengeling[9], Kathrin Maria Bernt[6,10], Philippa Marrack[1,2], John Kappler[1,2], Qiang Zhou[11], Chuan-Yuan Li[8], Yuhua Xue[4], Kirk Hansen[3], Gongyi Zhang[1,2]*

[1]Department of Biomedical Research, National Jewish Health, Denver, United States; [2]Department of Immunology and Microbiology, School of Medicine, University of Colorado, Aurora, United States; [3]Department of Genetics and Biochemistry, School of Medicine, University of Colorado, Aurora, United States; [4]State Key Laboratory of Cellular Stress Biology, School of Pharmaceutical Sciences, Xiamen University, Xiamen, China; [5]Department of Pediatrics, Children Hospital, University of Colorado, Aurora, United States; [6]Department of Pediatrics and the Center for Childhood Cancer Research, Children's Hospital of Philadelphia, Philadelphia, United States; [7]State Key Laboratory of Agrobiotechnology, China Agriculture University, Beijing, China; [8]Department of Dermatology, Duke University, Durham, United States; [9]Max-Planck-Society, Administrative Headquarters, Munich, Germany; [10]Department of Molecular and Cell Biology, University of California, Berkeley, United States; [11]Perelman School of Medicine, University of Pennsylvania, Philadelphia, United States

*For correspondence:
zhangg@njhealth.org

†These authors contributed equally to this work

**Abstract** More than 30% of genes in higher eukaryotes are regulated by promoter-proximal pausing of RNA polymerase II (Pol II). Phosphorylation of Pol II CTD by positive transcription elongation factor b (P-TEFb) is a necessary precursor event that enables productive transcription elongation. The exact mechanism on how the sequestered P-TEFb is released from the 7SK snRNP complex and recruited to Pol II CTD remains unknown. In this report, we utilize mouse and human models to reveal methylphosphate capping enzyme (MePCE), a core component of the 7SK snRNP complex, as the cognate substrate for Jumonji domain-containing 6 (JMJD6)'s novel proteolytic function. Our evidences consist of a crystal structure of JMJD6 bound to methyl-arginine, enzymatic assays of JMJD6 cleaving MePCE in vivo and in vitro, binding assays, and downstream effects of *Jmjd6* knockout and overexpression on Pol II CTD phosphorylation. We propose that JMJD6 assists bromodomain containing 4 (BRD4) to recruit P-TEFb to Pol II CTD by disrupting the 7SK snRNP complex.

## Introduction

Mechanisms of transcription regulation in bacteria are very well established; transcription factors bind to specific DNA to recruit RNA Polymerases (RNAP) to carry out transcription (*Ptashne and Gann, 1997*; *Zhang et al., 1999*). In eukaryotes, however, there are additional layers of regulation such as the nucleosome structures, which can prevent RNA Polymerases including RNA Polymerase I (Pol I), RNA Polymerase II (Pol II), and RNA Polymerase III (Pol III) from productive transcription due

**eLife digest** In animals, an enzyme known as RNA polymerase II (Pol II for short) is a key element of the transcription process, whereby the genetic information contained in DNA is turned into messenger RNA molecules in the cells, which can then be translated to proteins. To perform this task, Pol II needs to be activated by a complex of proteins called P-TEFb; however, P-TEFb is usually found in an inactive form held by another group of proteins. Yet, it is unclear how P-TEFb is released and allowed to activate Pol II.

Scientists have speculated that another protein called JMJD6 (Jumonji domain-containing 6) is important for P-TEFb to activate Pol II. Various roles for JMJD6 have been proposed, but its exact purpose remains unclear. Recently, two enzymes closely related to JMJD6 were found to be able to make precise cuts in other proteins; Lee, Liu et al. therefore wanted to test whether this is also true of JMJD6. Experiments using purified JMJD6 showed that it could make a cut in an enzyme called MePCE, which belongs to the group of proteins that hold P-TEFb in its inactive form.

Lee, Liu et al. then tested the relationships between these proteins in living human and mouse cells. The levels of activated Pol II were lower in cells without JMJD6 and higher in those without MePCE. Together, the results suggest that JMJD6 cuts MePCE to release P-TEFb, which then activates Pol II. JMJD6 appears to know where to cut by following a specific pattern of elements in the structure of MePCE. When MePCE was mutated so that the pattern changed, JMJD6 was unable to cut it. These results suggest that JMJD6 and related enzymes belong to a new family of proteases, the molecular scissors that can cleave other proteins.

The molecules that regulate transcription often are major drug targets, for example in the fight against cancer. Ultimately, understanding the role of JMJD6 might help to identify new avenues for cancer drug development.

to high binding affinity between DNA and histones. Precisely how RNA Polymerases overcome nucleosomal barriers to undergo a productive transcription elongation and how Pol II pausing is regulated remain unanswered (*Zhou et al., 2012*; *Jonkers and Lis, 2015*; *Core and Adelman, 2019*). In higher eukaryotes, over ~30% genes are regulated by Pol II promoter-proximal pausing (*Core et al., 2008*; *Nechaev et al., 2010*; *Min et al., 2011*), which is resultant from nucleosome barriers at +1 position of transcription start sites (*Weber et al., 2014*; *Voong et al., 2016*). We recently discovered that a group of JmjC domain containing protein family including JMJD5 and JMJD7 specifically cleave histone tails and potentially generate tailless nucleosomes. The cleavage activity by JMJD5 and JMJD7 could be associated with the release of the promoter-proximal paused Pol II and may trigger Pol II into productive elongation in higher eukaryotes, such as mouse and human (*Liu et al., 2017*; *Liu et al., 2018*). The cleavage activity of JMJD5 on histone tails was also independently reported by another group (*Shen et al., 2017*), thus cross-validating our respective discoveries.

Compared to the efficient recruitment of P-TEFb (including CDK9 and Cyclin T1) by TAT protein in human immunodeficiency virus (HIV) (*Peterlin et al., 2012*), BRD4 is claimed to be responsible for the recruitment P-TEFb to the promoters of Pol II pausing regulated genes (*Jang et al., 2005*; *Yang et al., 2005*; *C Quaresma et al., 2016*). However, the binding affinity between BRD4 and P-TEFb (~0.5 µM) (*Itzen et al., 2014*) is much weaker than that that of TAT and P-TEFb (~3 nM) (*Wei et al., 1998*; *Tahirov et al., 2010*; *Schulze-Gahmen et al., 2014*), and BRD4 lacks a RNA binding motif (*Wu and Chiang, 2007*). Therefore, we hypothesize there must exist another factor to help BRD4 to recruit P-TEFb and engages in the instigation of Pol II transcription elongation. Besides the classic Bromo-domains which recognize acetylated histone tails, BRD4 contains an extra terminal domain (ET) recognizing JMJD6 (*Rahman et al., 2011*; *Konuma et al., 2017*). Incidentally, we found that JMJD6 nonspecifically binds to single stranded RNA with high affinity (~40 nM) (*Hong et al., 2010*). We propose that JMJD6 may be recruited by both BRD4 and newly transcribed RNAs from Pol II to help BRD4 recruit P-TEFb, acting analogously to that of TAT protein associating with both P-TEFb and TAR.

JMJD6 is one of the most controversial proteins in biology (*Vangimalla et al., 2017*). It was first cloned as phosphatidylserine (PS) receptor (*Fadok et al., 2000*), but was corrected as a nucleus expressed protein unrelated to PS (*Böse et al., 2004*; *Cikala et al., 2004*; *Cui et al., 2004*). It was

later reported to contain arginine demethylase activity on histone tails (*Chang et al., 2007*), hydroxylase activity on splicing factor U2AF65 (*Webby et al., 2009*) and histone tails (*Han et al., 2012*), and both arginine demethylase activities on histone tails and RNA demethylase activities on 5' prime of 7SK snRNA (*Liu et al., 2013*), and surprisingly PS binding (*Neumann et al., 2015*; *Yang et al., 2015*). The exact or cognate substrate(s) of JMJD6 remains unresolved or controversial. Based on the novel protease activities of JMJD5 and JMJD7 (*Liu et al., 2017*; *Shen et al., 2017*; *Liu et al., 2018*), the high structural similarity among catalytic cores of JMJD5, JMJD6, and JMJD7 (*Hong et al., 2010*; *Liu et al., 2018*), and analogous severe phenotypes among knockouts of *Jmjd5* and *Jmjd6* in mice (*Li et al., 2003*; *Böse et al., 2004*; *Ishimura et al., 2012*; *Oh and Janknecht, 2012*), we hypothesized that JMJD6 may contain protease activity working on methylated arginines on some protein candidates which regulate the activity of Pol II, especially promoter-proximally paused Pol II.

It is well established that the 7SK snRNP complex primarily functions to sequester the CDK9-containing P-TEFb until stimulation (*Jang et al., 2005*; *Yang et al., 2005*). MePCE (methylphosphate capping enzyme) was first characterized as a component of the 7SK snRNP complex which acts as a capping enzyme on the gamma phosphate at the 5'end of 7SK RNA (*Jeronimo et al., 2007*). Furthermore, a capping-independent function of MePCE via stabilization of 7SK snRNA and facilitation in the assembly of 7SK snRNP was reported by Dr. Qiang Zhou's group (*Xue et al., 2010*). Knockdown of MePCE led to destabilization of the 7SK snRNP complex in vivo (*Xue et al., 2010*; *Singh et al., 2011*; *C Quaresma et al., 2016*). A nonsense variant of MePCE is reported to be associated with a neurodevelopmental disorder exhibiting hyperphosphorylation of Pol II, potentially caused by enhanced activation of CDK9 complex (*Schneeberger et al., 2019*). Interestingly, one report showed that MePCE may also work in an 7SK snRNP independent manner to recruit CDK9 on a small group of genes (*Shelton et al., 2018*). In this report, we reveal that MePCE of the 7SK snRNP complex is a cognate substrate of JMJD6.

## Results

### JMJD6 has a unique structure to hold methyl-arginine

Based on these divergent reports regarding substrates of JMJD6 (*Chang et al., 2007*; *Webby et al., 2009*; *Han et al., 2012*; *Liu et al., 2013*; *Neumann et al., 2015*), we re-interrogated proposed substrates using stringent and unified criteria. As we reported previously, JMJD6 binds with high binding affinity (~40 nM) to single stranded RNA (ssRNA) without sequence specificity (*Hong et al., 2010*). However, truncation analysis showed that JMJD6 barely binds to ssRNA without the C-terminal flexible region (*Hong et al., 2010*). This suggests that the C-terminal domain of JMJD6 may just serve as ssRNA binding motif and RNAs are not a substrate for the enzymatic activity of JMJD6. On the other hand, the structure of the catalytic core of JMJD6 shows some critical similarity to those of JMJD5 and JMJD7, with a negatively charged microenvironment near the catalytic center (*Hong et al., 2010*; *Liu et al., 2018*), suggesting positively charged substrates (*Figure 1*). As we reported, JMJD5 and JMJD7 specifically recognize methylarginines of histone tails *via* a Tudor-domain-like structure near the catalytic center of JMJD5, which could specifically recognize methylarginines, but not methyllysine (*Liu et al., 2017*; *Liu et al., 2018*). We reasoned that the similar structural features among JMJD6, JMJD5, and JMJD7 may confer a similar substrate for JMJD6 as those of JMJD5 and JMJD7. In this regard, crystals of JMJD6 without C-terminal motif (1-343) were soaked with a monomethylarginine derivative. Interestingly, four out of eight JMJD6 molecules within an asymmetric unit bound to monomethylarginine, which coordinates with $Fe^{2+}$ and alpha-KG in the catalytic center similar to that of JMJD5 and methylarginines (*Figure 1*, *Figure 1—figure supplements 1–3*, *Supplementary file 1*). However, the methylated sidechain of arginine is located in a more open catalytic space containing negatively charged residues, compared to that of JMJD5, indicating that the pocket could hold more than one sidechain (*Figure 1A*). This may suggest a novel substrate recognition mode, which is different from that of JMJD5 (*Liu et al., 2018*). Nevertheless, the complex structure shows several key evidences. First, JMJD6 does bind to substrates with methylarginine or possibly methyllysine or both arginine and lysine with and without methylation (*Figure 1B,C*). Second, the methyl group is far away from either the divalent ion or alpha-KG, suggesting JMJD6 may not act as lysine or arginine demethylases to remove methyl groups on the

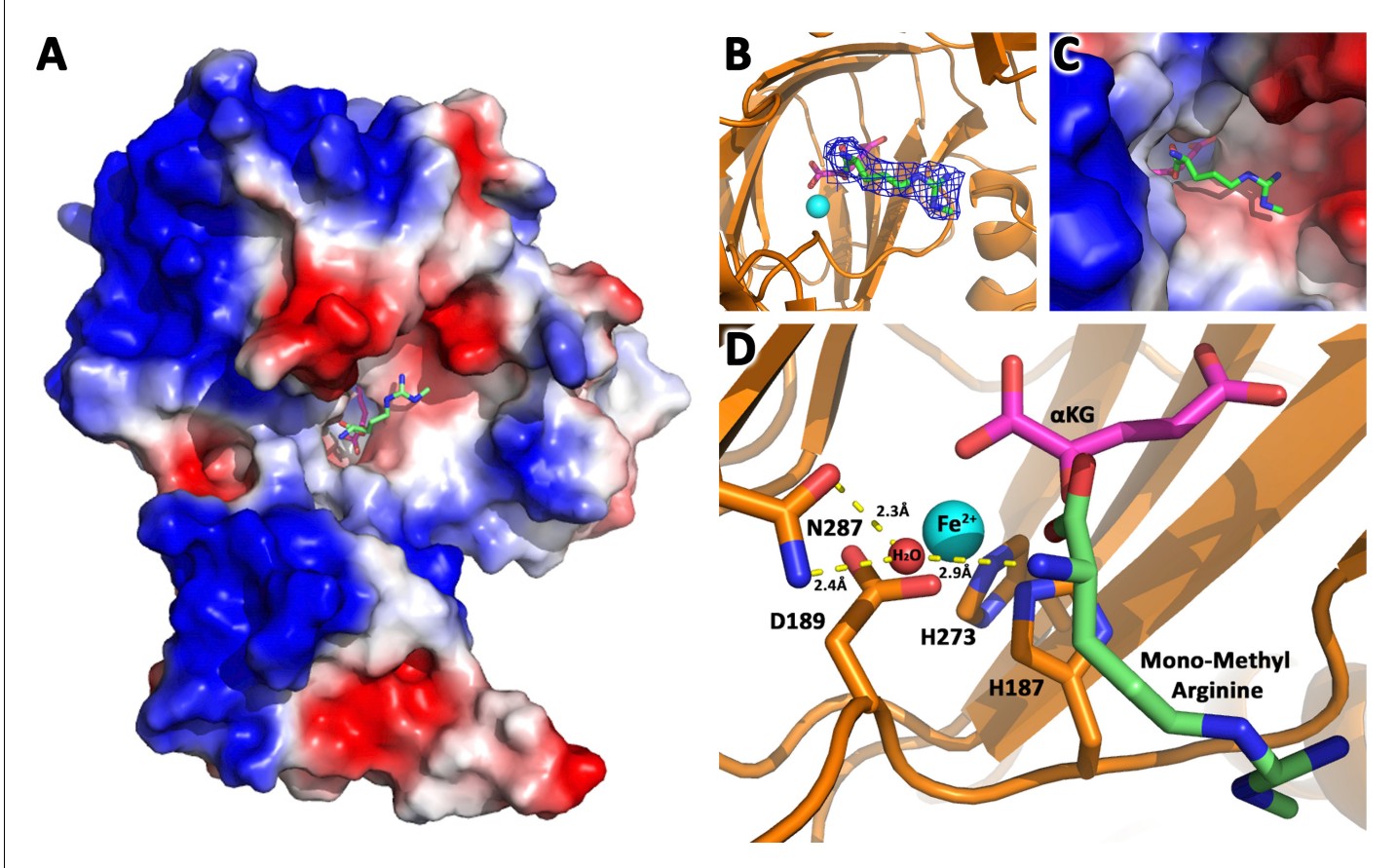

**Figure 1.** JMJD6 binds to monomethyl arginine (1-of-4). (A) Complex structure of JMJD6 (1–343) and monomethyl arginine (MM-Arg). Surface charges were generated using PyMOL (Action > generate > vacuum electrostatics > protein contact potential; https://pymol.org/2/). Red represents negatively-charged surface, Gray represents neutral-charged surface, and Blue represents positively-charged surface. (B) Omit map 2Fo-Fc electron density of MM-Arg. (C) Magnified view of MM-Arg in the catalytic center of JMJD6 (D) Coordination of elements at catalytic center.
The online version of this article includes the following figure supplement(s) for figure 1:

**Figure supplement 1.** JMJD6 binds to monomethyl arginine (2-of-4).
**Figure supplement 2.** JMJD6 binds to monomethyl arginine (3-of-4).
**Figure supplement 3.** JMJD6 binds to monomethyl arginine (4-of-4).

sidechain of either lysine or arginine. Third, peptides or proteins could be cognate substrates instead of RNA or DNA, which do not contain any positive charge with or without methylation. Fourth, the catalytic center contains analogous residues present in JMJD5 and JMJD7, suggesting a similar novel catalytic mechanism as those of JMJD5 and JMJD7 through an imidic acid as proton mediator (*Figure 1D*; *Lee et al., 2017*; *Liu et al., 2018*). This novelty is particularly exemplified by the fact that commercially available protease inhibitor cocktail (Roche) at 2x concentration cannot inhibit activities of JMJD5, JMJD6, and JMJD7. Furthermore, a comprehensive protein composition analysis using mass spectrometry (*Supplementary file 2*) of all purified recombinant JMJD6 samples used in the following experiments could not detect any protease contaminants.

## JMJD6 cleaves MePCE

Since JMJD5 and JMJD7 make cleavage on histone tails (*Liu et al., 2017*), we asked whether JMJD6 also recognizes histone tails. Interestingly, JMJD6 does in fact have activities on bulk histone in vitro (*Figure 2—figure supplement 1A*). This result may explain why two groups found that JMJD6 could reduce methylarginine containing histone tails in in vitro assays probed with antibodies (*Chang et al., 2007*; *Liu et al., 2013*), in which post-cleft short peptides with methylarginines cannot be recognized by antibodies in western blots. To confirm or rule out whether histone tails are

cognate substrates of JMJD6, we first assessed the binding affinity between JMJD6 and histone peptides with or without arginine methylation. From fluorescence polarization binding assays, binding affinities between JMJD6 and peptides are weak and around ~150 µM (*Figure 3—figure supplement 1D,E*). Most importantly, with or without *Jmjd6*, the level of arginine methylated histones or overall histone levels does not change in MEF cells in vivo (*Figure 2—figure supplement 1B*), which is in stark contrast to those of JMJD5 and JMJD7 (*Liu et al., 2017*). These data suggest that histone tails are not cognate substrates of JMJD6. Parenthetically, it is reported that JMJD6 binds to LARP7 on 7SK snRNP (*Weimann et al., 2013*), works on methyl cap of 5' 7SK snRNA (*Liu et al., 2013*), and binds to BRD4 (*Rahman et al., 2011*; *Konuma et al., 2017*), all suggesting a close relation with the 7SK snRNP complex.

We hypothesized that JMJD6 may work on some protein component(s) of the 7SK snRNP to regulate the stability of 7SK snRNP complex. Interestingly, all protein components of the 7SK snRNP complex including MEPCE, LARP7, and HEXIM1 are drastically decreased in the MEF cells without *Jmjd6* (*Figure 2A*). One possibility is that all members are transcriptionally regulated by JMJD6. This could be consistent with the severe phenotype of *Jmjd6* knockouts, which results in neonatal lethality and cell growth retardation and tissue differentiation (*Li et al., 2003*; *Böse et al., 2004*), suggesting that JMJD6 is a global master transcriptional regulator controlling expression of a large group of genes including components from the 7SK snRNP complex. However, our RNA-Seq data does not support direct transcriptional downregulation of 7SK snRNP complex members at mRNA level (*Supplementary file 3*). All of them, including MePCE, LARP7, and HEXIM1 have similar mRNA levels with or without *Jmjd6*. At the moment, the cause of this downregulation of proteins level of these components from 7SK snRNP remains unresolved and is beyond the scope of this report. However, when we introduce back JMJD6 via overexpression into MEF cells lacking *Jmjd6*, protein levels of LARP7 and HEXIM1 are rescued, whereas MePCE is undetectable (*Figure 2A*) or significantly reduced (*Figure 2—figure supplement 1C*). To account for this disappearance of MePCE, we hypothesized that overexpression of JMJD6 may directly target MEPCE for degradation. To confirm this hypothesis, we respectively overexpressed full-length MePCE in wild-type MEF, *Jmjd6* knockout MEF, wild-type JMJD6 overexpression in *Jmjd6* knockout MEF, and inactive mutant JMJD6 overexpression in *Jmjd6* knockout MEF. The whole cell lysates were probed with anti-MePCE antibody. Our hypothesis was vindicated with the emergence of a lower molecular weight form of MePCE in wild-type MEF and wild-type JMJD6 overexpression in *Jmjd6* knockout MEF, whereas the lower molecular weight form of MePCE was not detectable in the *Jmjd6* knockout MEF, and inactive mutant JMJD6 overexpression in *Jmjd6* knockout MEF (*Figure 2B*).

To reproduce these in vivo results in an in vitro setting, full-length proteins linked with N-terminal His[6]-tag of MePCE and HEXIM1 were generated from insect cells and LARP7 and HEXIM2 were generated from bacteria and purified, followed by subjecting these proteins to direct in vitro enzymatic assays. Consistent with our in vivo results, JMJD6 cleaves MePCE in vitro (*Figure 2C,D*), but not HEXIM1, HEXIM2, nor LARP7 (data not shown). Upon cleavage of MePCE by JMJD6 in vitro, a band with molecular weight ~25 kDa was detected by antibody against His[6]-tag at the N-terminal (*Figure 2C*). To provide context, MePCE contains a total of 689 residues and a theoretical molecular weight of 74.4 kDa, whereas it is detected at ~90 kDa on SDS-PAGE gel due to its high content of proline residues (10.7%). We expect that the ~25 kDa fragment from N-terminal of MePCE may contain 200 residues or less due to rich proline residues within the N-terminal of MePCE (first 150 residues of MePCE contains 27 prolines, 18.0% prolines). Furthermore, our in vivo assay yielded a cleaved MePCE band with a molecular weight ~65 kDa (*Figure 2B*). The anti-MePCE antibody used in this assay was generated using a peptide immunogen containing residues 200–250 of MePCE. Thus, the ~65 kDa band may correspond to the cleaved C-terminal segment of MePCE containing the approximate residues 200–689.

Based on the binding of JMJD6 to methylated arginine we obtained from the complex crystal structure, we reasoned that arginine residues within approximately the first 200 residues of MePCE could contain the recognition site. Several peptide fragments including residues from 81 to 160, residues 154 to 184, and residues 187 to 244 were synthesized. Peptides of 81–160 and 187–244 did not show any cleavage when incubated with JMJD6. Peptides of 154–184 showed cleavage activity, but at levels of <1% compared to peptide input. We attributed this low activity as a matter of insolubility and dimerization via C177 oxidation. To overcome this obstacle, a shorter peptide from residue 161 to 179 containing C177S was synthesized and subjected to enzymatic reaction under

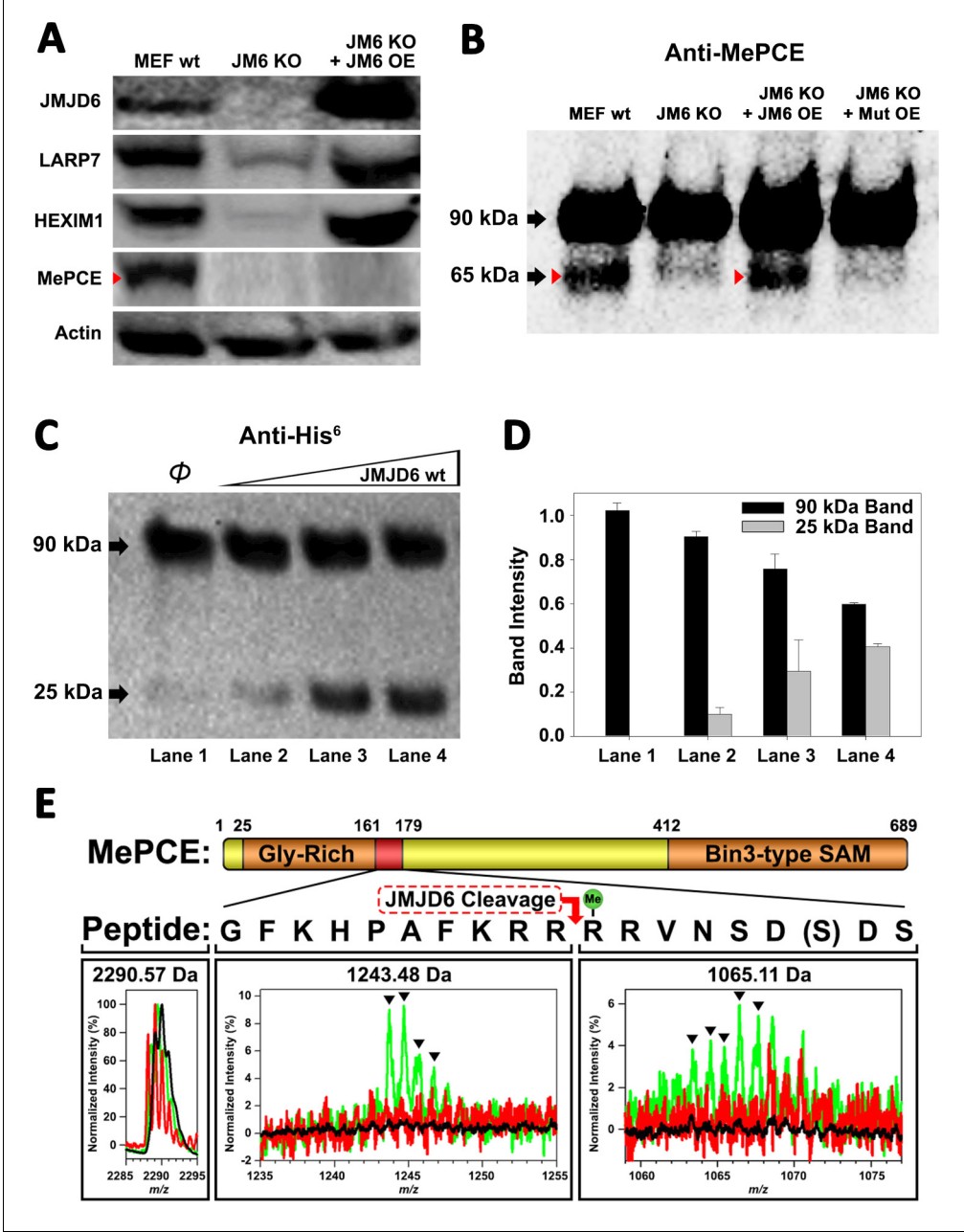

**Figure 2.** JMJD6 targets MePCE for proteolysis. (**A**) Western blot of wild-type MEF, *Jmjd6* knockout MEF, and JMJD6 overexpression in *Jmjd6* knockout MEF probed with antibodies specific for JMJD6, Actin, and components of the 7SK snRNP complex; LARP7, HEXIM1, and MePCE. (**B**) Western blot of MePCE overexpressed respectively in wild-type MEF, *Jmjd6* knockout MEF, wild-type JMJD6 overexpression in *Jmjd6* knockout MEF, and inactive mutant JMJD6 overexpression in *Jmjd6* knockout MEF; probed with antibody specific for MePCE. (**C**) Wild-type JMJD6 titrated into full-length MePCE with N-terminal His[6]-tag. Enzymatic activity of JMJD6 is probed with anti-His[6] antibody. (**D**) Quantification of c. (**E**) The endopeptidase activity of JMJD6 on synthesized MePCE (161–179) R171-me2s/C177S peptide. The mass spectrum is normalized to the intensity of the undigested peptide input. The peptide is assayed with wild-type JMJD6 (green), inactive mutant JMJD6 (red), or peptide alone (black). The MePCE (161–179) peptide with symmetric dimethylation on R171 and C177S mutation has a molecular weight of 2,290.57 Da. After wild-type JMJD6 cleavage between R170 and R171, the major peaks* (black triangles) with the molecular weight of 1,243.48 Da corresponds to the N-terminal product and the molecular weight of 1,065.11 Da corresponds to the C-terminal product respectively. *The multiple peaks are isotopic distributions, which are characteristic of MALDI-TOF.

*Figure 2 continued on next page*

*Figure 2 continued*
The online version of this article includes the following figure supplement(s) for figure 2:
**Figure supplement 1.** Histone tails are not the cognate substrate of JMJD6.

JMJD6. This peptide also contains symmetric dimethylation on R171, given that our binding data, described in the next section, suggests that this particular modification yields the highest binding affinity (*Figure 3C*). In line with our expectations, dominant peaks corresponding to the cleaved MePCE peptide products were detected by mass spectrometry, but not in either peptide alone or with an inactive mutant version of JMJD6 (*Figure 2E*).

## JMJD6 specifically recognizes methylarginine within MEPCE

To determine the binding affinity between JMJD6 and the newly discerned MePCE proteolysis site, Microscale Thermophoresis (MST) assay was performed using catalytic core of JMJD6 (1–343) titrated with the MePCE (154-184) peptide exhibiting: no modification (*Figure 3A*), R170-me2s (*Figure 3B*), or R171-me2s (*Figure 3C*). The highest binding affinity was exhibited by the MePCE (154-184)-R171-me2s peptide with Kd = 159 ± 77 nM, followed by the peptide containing no modification with Kd = 471 ± 292 nM, and lastly the peptide with R170-me2s with Kd = 568 ± 316 nM. Fluorescence polarization assays were used to cross-validate the Kd values obtained via MST above, all of which respectively fell within the margin of error (*Figure 3—figure supplement 1A,B,C*). These data suggest that methylation on R171 enhances binding nearly 3-fold compared to no modification, whereas methylation on R170 diminishes binding marginally. Whether or not these particular methylations occur in vivo is yet to be explored. Interestingly, the YASARA Energy Minimization Server developed by Dr. Kevin Karplus' group yielded a theoretical model that positions R171 in the identical position to the monomethyl-arginine observed in our crystal model (*Figure 3—figure supplement 2A*; *Krieger et al., 2009*). Furthermore, the energy-minimized model displayed eight separate charge-charge interactions between JMJD6 and MePCE (*Figure 3G*). Although a *bona fide* complex structure is preferred, this computational model is in excellent agreement with all our findings and provides a coherent justification for the experimentally observed high binding affinities between JMJD6 and MePCE. To compare the abovementioned binding affinities to that of JMJD6 and histone tails, as other groups have previously purported, the identical MST assay was performed using peptides derived from Histone 3 (1-21) containing no modification (H3) (*Figure 3D*), Histone 3 (1-16)-R2-me2s (H3R2me2s) (*Figure 3E*), and C-peptide (57-87) as a negative control (*Figure 3F*). For the Histone three peptides, low binding affinity was exhibited with Kd = 77 ± 34 μM for H3 peptide and Kd = 98 ± 14 μM for H3R2me2s peptide. No meaningful binding was observed for the C-peptide (*Figure 3F*). To verify the authenticity of the above model and confirm our proposed cleavage site of JMJD6 within MePCE, we generated an MePCE knockout 293 T cell line (*Figure 3H*) and respectively overexpressed C-terminal His[6]-tagged wild-type MePCE or C-terminal His[6]-tagged mutant MePCE, which replaced -RRRR- to -AAAA- at the cleavage site. Interestingly, when probed with anti-His[6] antibody in a western blot, wild-type MePCE generates two bands of MePCE (*Figure 3I*, left lane), consistent with *Figure 2B*, whereas the mutant version does not (*Figure 3I*, right lane), suggesting that this site is the authentic cleavage site for JMJD6. To further confirm these results, we introduced the two variants of MePCE back to the JMJD6 overexpressing MEF cell line in *jmjd6* knockout background. Once again consistent with *Figure 2B*, two major bands show up with the wild-type MePCE (*Figure 3J*, left lane), whereas only one major band and one faint band show up with the mutant MePCE (*Figure 3J*, right lane). This data provides another solid evidence of the cleavage site within MePCE by JMJD6. To further verify our proposed role of JMJD6 on MePCE, the mutant MePCE containing N-terminal His[6]-tagged was expressed in insect cells and purified for an in vitro assay. The native form of MePCE generates a small band (*Figure 3—figure supplement 2B*), consistent with *Figure 2C*, whereas the mutant MePCE does not generate any additional band (*Figure 3—figure supplement 2C*). Taken together, we propose that JMJD6 is the cognate protease of MePCE and cleaves at the R171 site within MePCE.

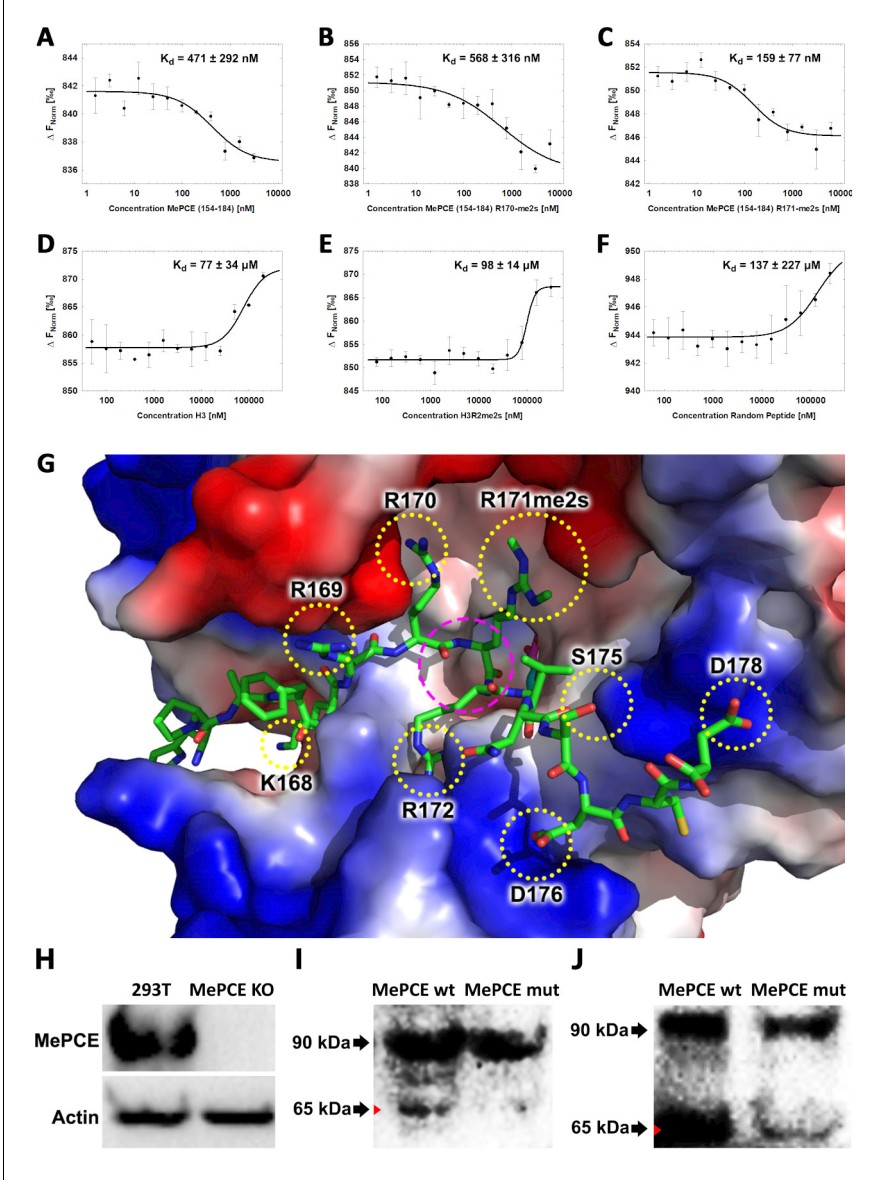

**Figure 3.** JMJD6 specifically binds to MePCE site containing residues 164–178 (determined via MST). (**A**) The binding of His-JMJD6 (1–343) to unmodified MePCE (154-184). (**B**) The binding of His-JMJD6 (1–343) to MePCE (154-184) R170-me2s. (**C**) The binding of His-JMJD6 (1–343) to MePCE (154-184) R171-me2s. (**D**) The binding of His-JMJD6 (1–343) to unmodified Histone 3 (1-21). (**E**) The binding of His-JMJD6 (1–343) to Histone 3 (1-16) R2-me2s. (**F**) The binding of His-JMJD6 (1–343) to C-peptide (57-87). (**G**) Electrostatic interactions between JMJD6 (1–343) and MePCE (164-178) determined by YASARA Energy Minimization server are highlighted in yellow dotted circles and catalytic center is highlighted in magenta dashed circle. (**H**) Western blot of wild-type 293 T cells (left lane) and MePCE knockout 293 T cells (right lane) probed with antibody specific for MePCE. (**I**) Western blot of MePCE knockout 293 T cells overexpressing C-terminal His[6]-tagged wild-type MePCE (left lane) or C-terminal His[6]-tagged mutant MePCE (right lane), respectively, probed with antibody specific for His[6]-tag. (**J**) Western blot of wild-type JMJD6 overexpression in *Jmjd6* knockout MEF cells overexpressing C-terminal His[6]-tagged wild-type MePCE (left lane) or C-terminal His[6]-tagged mutant MePCE (right lane), respectively, probed with antibody specific for His[6]-tag.

The online version of this article includes the following figure supplement(s) for figure 3:

**Figure supplement 1.** JMJD6 specifically binds to MePCE site containing residues 164–178 (determined via fluorescence polarization).

**Figure supplement 2.** Computational complex structure model of JMJD6 (1–343) and MePCE (164-178) derived from YASARA Energy Minimization server and verification.

## Knockout of *Jmjd6* leads to down-regulation of Ser2p-CTD of pol II

We previously reported that cleavage of histone tails by JMJD5 and JMJD7 could lead to tailless nucleosomes, which are unstable and are more readily overcome by Pol II during transcription (*Liu et al., 2017*; *Liu et al., 2018*). What happens when MePCE is cleaved? Does MePCE cleavage mediate P-TEFb to be released from the 7SK snRNP complex, allowing P-TEFb to be recruited to CTD of Pol II, and then to phosphorylate Ser2-CTD of Pol II? Nuclear extracts from both wild-type MEFs, *Jmjd6* knockout MEFs and JMJD6 overexpression in *Jmjd6*-deficient MEFs were subjected to investigation as to the level of Ser2p-CTD of Pol II. Consistent with all our current and previous findings, Ser2p-CTD of Pol II is drastically reduced (~80%) in MEF cells without *Jmjd6*, despite CDK9 levels remaining unchanged, suggesting that Ser2p-CTD of Pol II is regulated by JMJD6 (*Figure 4A,B*). A question raised here is whether the drop of active Pol II (Ser2p-CTD) is related to the protease activity of JMJD6 toward the CDK9-sequestering role of MePCE. Therefore, the MePCE knockout cell line was investigated for its effect on Pol II. To our great satisfaction, Pol II Ser2p-CTD is dramatically increased in 293 T cells with MePCE knockout (*Figure 4C*, right lane) compared to the wild-type (*Figure 4C*, left lane). Of interest, *Shelton et al. (2018)* showed that CDK9 interacts with the N-terminal region of MePCE (1-400)). We found a very similar JMJD6 cutting site around residues 363–367 (-RKRRR-) compared the site we characterized around residues 167–172 (-KRRRR-). If JMJD6 also cuts this site as well, it seems that MePCE may act as an anchor site for CDK9 on 7SK snRNP complex. It is an interesting topic that requires further investigation.

The totality of our results is consistent with the critical role of JMJD6 in embryonic development, which is neonatal lethal without *Jmjd6* (*Li et al., 2003*; *Böse et al., 2004*). It explains why growth of MEF cells without *Jmjd6* is severely compromised, similar to those without either *Jmjd5* or *Jmjd7* (*Liu et al., 2017*). This data suggests not only that JMJD6's cleavage of MePCE is essential for P-TEFb to be released from the 7SK snRNP complex and ultimately the phosphorylation of Ser2-CTD of Pol II, but also further cross-confirms that a species of Ser2p-CTD of Pol II is indeed generated by CDK9. The content of Ser2p-CTD of Pol II is recovered when JMJD6 is re-expressed (*Figure 4A,B*), suggesting the direct relation of JMJD6 and phosphorylation of Ser2-CTD of Pol II in vivo. When these results are aggregated with previously published reports, whereby (1) JMJD6 exhibits strong binding affinity to nonspecific ssRNA as we previously reported (*Figure 4D*; *Hong et al., 2010*), (2) JMJD6 associates with BRD4 as reported by others (*Figure 4D*; *Rahman et al., 2011*; *Konuma et al., 2017*), and (3) JMJD6 digests MePCE so as to disrupt the overall stability of 7SK snRNP complex as highlighted in this report (*Figure 4D*), a model of P-TEFb transcription regulation unique to higher eukaryotes can be generated: JMJD6 binds to 20–50 nt newly transcribed 5' prime end of ssRNAs of initiated Pol II of any stimulating genes. The association between BRD4 and JMJD6 allows BRD4 to bring P-TEFb (CDK9) to close proximity of CTD of Pol II, allowing for Ser2-CTD phosphorylation (*Figure 4E*). Remarkably, this recruitment of P-TEFb by JMJD6, with the assistance of BRD4, is similar to that of TAT protein hijacking P-TEFb complex after associating with TAR transcript to trigger the expression of HIV retroviral genome (*Figure 4F*; *Peterlin et al., 2012*; *Zhou et al., 2012*; *C Quaresma et al., 2016*). Here, our observation regarding the direct relation of JMJD6 and level of Ser2p-CTD of Pol II in vivo is corroborated by Liu et al.'s report, in which 7SK snRNP complex pulled-down through HEXIM1 antibody from HEK293 cells could be disrupted by JMJD6 in vitro, as indicated by the decrease in CDK9 levels (*Liu et al., 2013*), thus strongly supporting our proposed role of JMJD6 as a direct disruptor of the 7SK snRNP complex.

## Discussion

It took us nearly two decades to address the structure and function of JMJD6 since it was first cloned (*Fadok et al., 2000*). During the long journey, based on the conserved structure of JmjC domain containing family, we pioneered in characterizing the catalytic core of JMJD2 subfamily, which turned out to be a novel lysine demethylase (*Chen et al., 2006*; *Whetstine et al., 2006*; *Chen et al., 2007*). We then solved the structure of JMJD6 with the help of a monoclonal antibody and discovered its unique ssRNA binding property almost a decade ago (*Hong et al., 2010*). However, the exact function and its cognate substrate remained a mystery, though several groups claimed variety of enzymatic activities and substrates in past two decades (*Chang et al., 2007*; *Webby et al., 2009*; *Han et al., 2012*; *Liu et al., 2013*). The accidental discoveries of novel protease

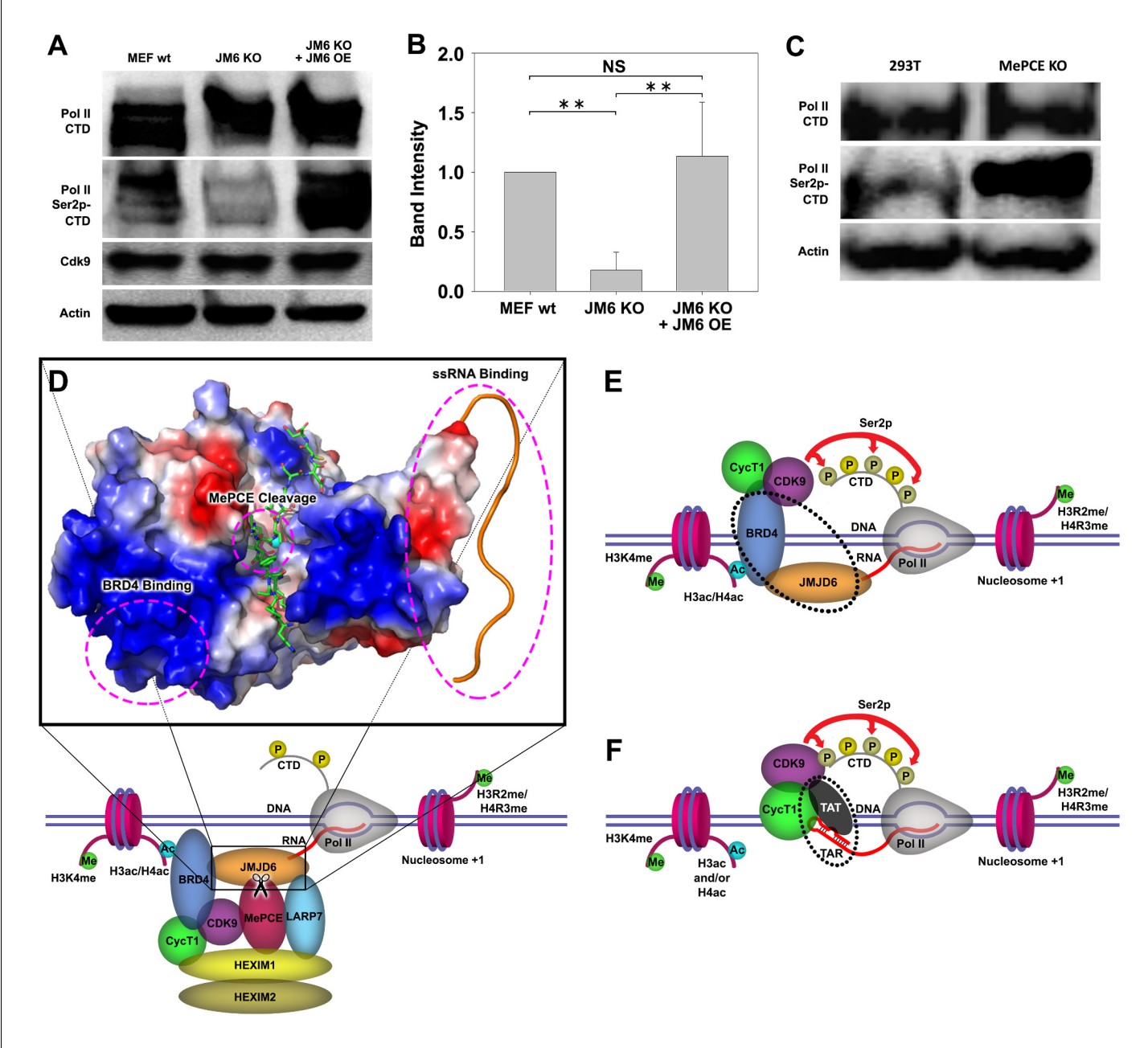

**Figure 4.** JMJD6 regulates Pol II Ser2-CTD phosphorylation. (A) Western blot of wild-type MEF, *Jmjd6* knockout MEF, and JMJD6 overexpression in *Jmjd6* knockout MEF probed with antibodies specific for Pol II CTD, Pol II ser2p-CTD, CDK9, and Actin. (B) Quantification of Pol II Ser2p-CTD on a. (**, p<0.01; NS, not significant) (C) Western blot of wild-type 293 T cells and *MePCE* knockout 293 T cells probed with antibodies specific for for Pol II CTD, Pol II ser2p-CTD, and Actin. (D) Model of JMJD6 cleavage of MePCE within the 7SK snRNP complex. A representative surface charge model of JMJD6 with unstructured C-terminal tail highlights respective JMJD6 interaction sites with MePCE, BRD4, and ssRNA in magenta dashed circles. (E) ssRNA-bound JMJD6 and acetylated H3/H4-bound BRD4 in conjunction (black dotted circle) bridges P-TEFb to paused Pol II. (F) TAR-bound TAT (black dotted circle) bridges P-TEFb to paused Pol II.

activities of JMJD5 and JMJD7 by our group prompted the exploration of potential protease activity of JMJD6 (*Liu et al., 2017*; *Liu et al., 2018*). The report by *Liu et al. (2013)* helped us to narrow down the potential cognate substrate within the 7SK snRNP complex. Reports of arginine demethylase activity (*Chang et al., 2007*; *Liu et al., 2013*) helped us to focus on methylated arginine as the potential cleavage sites. Thanks to these reports, we identified MePCE to be the cognate substrate

of JMJD6, as well as the potential cleavage site. Numerous lines of evidence from our current discoveries and other publications corroborates the authenticity of this conclusion: First, JMJD6 is specifically associated with BRD4 through the ET domain (*Rahman et al., 2011*; *Konuma et al., 2017*), while BRD4 is very well established to recruit P-TEFb to paused Pol II (*Yang et al., 2005*; *Yang et al., 2008*). Second, JMJD6 could disrupt the 7SK snRNP complex in vitro as reported by *Liu et al. (2013)*. Third, our current discoveries showed that JMJD6 cleaves MePCE of the 7SK snRNP complex. Fourth, our previous discovery of JMJD6's interaction with non-specific ssRNA bridges JMJD6 to the initiated Pol II which generates 20–50 nt long ssRNA (*Hong et al., 2010*).

The findings that JMJD5 and JMJD7 are in fact proteases was highly unexpected, and required multiple lines of evidence pursued with great scrutiny. It is quite extraordinary that a very conserved Jumonji domain containing hydroxylase family should contain a subfamily which evolutionarily adopted distinctive protease activities, as well as exhibit several unexpected novelties including the catalysis mechanism involving imidic acid and both endopeptidase and exopeptidase activities (*Liu et al., 2017*; *Liu et al., 2018*). Indeed, recombinant proteins derived from bacteria have a high chance of contamination by bacterial proteases, although we exhausted innumerable means to exclude the possibility of contamination in our assays (*Liu et al., 2017*; *Liu et al., 2018*). In this report, the novel protease activity of JMJD6, which cleaves *before* the methylated arginine, is remarkably distinct to the cleavage *after* the methylated arginine mediated by JMJD5 and JMJD7 (*Liu et al., 2017*; *Liu et al., 2018*). This important piece of evidence supports the claim of JMJD5 and JMJD7 as proteases. It is highly improbable that one batch of proteins we purified was contaminated by one type of undiscovered protease(s) (cleavage after methylated arginine for JMJD5 and JMJD7) while another batch of proteins was contaminated by a different type of undiscovered protease(s) (cleavage before methylated arginine for JMJD6), all which respectively have substrate specificity matching the expected biological profiles of JMJD5/6/7.

Based on our current discoveries, we may derive a novel transcription regulation pathway for genes regulated by promoter-proximal pausing Pol II. First, upon stimulation of cells, signals will reach specific transcription factors through signal transduction pathways. With modification or with the help of other partner molecules, these transcription factors bind to enhancers close to the paused Pol II. These transcription factors will recruit P300/CBP with the help of H3K4(me1) at enhancer regions, which in turn acetylates H3 and/or H4 on the same nucleosome bound by P300/CBP through association with H3K4(me1) to generate acetylated H3 and/or H4 (*Figure 5A*). Next, BRD4 is recruited to acetylated H3 and/or H4, and in turn engages with JMJD6 and the initiated Pol II complex. JMJD6 specifically cleaves MePCE of the 7SK snRNP complex so as to release the P-TEFb complex containing CDK9 (*Figure 5B*). Finally, JMJD6 and BRD4 brings P-TEFb to close proximity of CTD of Pol II through super elongation complex. P-TEFb (CDK9) then phosphorylates Ser2-CTD of Pol II (*Figure 5C*).

# Materials and methods

## Key resources table

| Reagent type (species) or resource | Designation | Source or reference | Identifiers | Additional information |
|---|---|---|---|---|
| Cell line (*M. musculus*) | JMJD6 KO cell on MEF background | *Böse et al., 2004* | | Cell line from Dr. Andreas Lengeling lab |
| Cell line (*M. musculus*) | JMJD6 Re-expression cell line on the JMJD6 KO background | This paper | | MSCV-JMJD6 transduced JMJD6 KO cell |
| Cell line (*Homo-sapiens*) | 27–13 MEPCE KO cell on the HEK293 background | This paper | | Cell line from Drs. Yuhua Xue lab |
| Cell line (*Homo-sapiens*) | MEPCE Re-expression cell line on the MEPCE KO cell line 27–13 background | This paper | | MSCV-MEPCE with c-terminal 6*His tag transduced MEPCE KO cell |
| Antibody | Rabbit polyclonal Phospho-Serine 2 CTD Pol II | Nat Struct Mol Biol 15, 71–78 (2008) | Dr. David Bentley Lab | WB (1:2000) |

*Continued on next page*

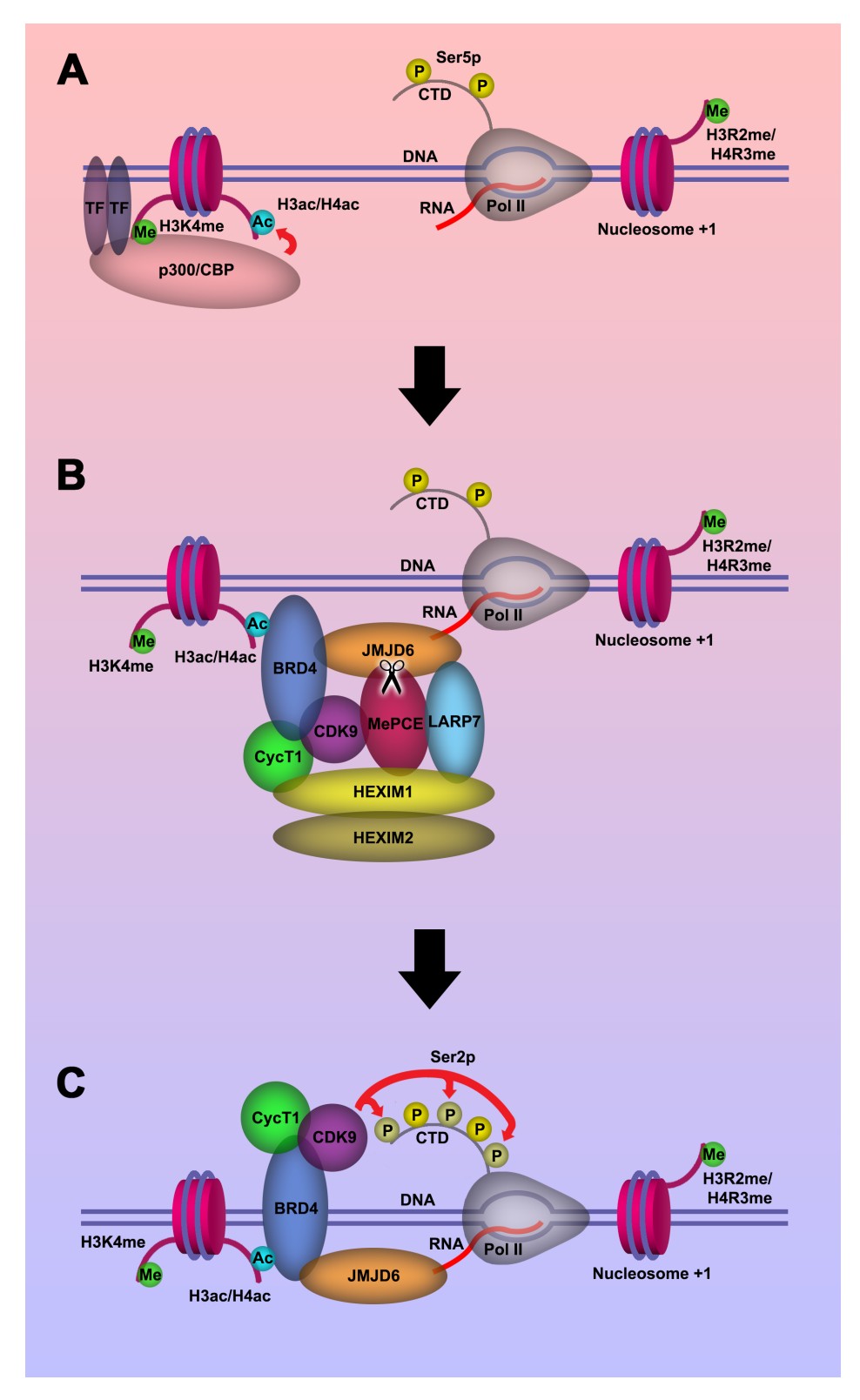

**Figure 5.** Model of P-TEFb release from 7SK snRNP complex. (a.) Pol II is initiated at the TSS and remains in the paused state until further stimulation. Transcription factors and H3K4me recruits p300/CBP, which acetylates and generates H3ac and/or H4ac. (b) H3ac and/or H4ac recruits BRD4, which associates with 7SK snRNP/P-TEFb complex and JMJD6. JMJD6 associates with ssRNA from Pol II. JMJD6 digests MePCE to disrupt the 7SK snRNP complex to release P-TEFb (CDK9). (c) BRD4/JMJD6 brings CDK9 in close proximity of CTD of Pol II. CDK9 phosphorylates Ser2 motifs on CTD of Pol II.

*Continued*

| Reagent type (species) or resource | Designation | Source or reference | Identifiers | Additional information |
|---|---|---|---|---|
| Antibody | Rabbit polyclonal CTD Pol II antibody | Nat Struct Mol Biol 15, 71–78 (2008) | Dr. David Bentley Lab | WB (1:2000) |
| Antibody | Mouse monoclonal JMJD6 antibody | Santa Cruz Biotechnology | sc-28348 | WB (1 μg/ml) |
| Antibody | Rabbit polyclonal MEPCE antibody | Bethyl Laboratories Inc | A304-184A | WB (1:2000) |
| Antibody | Rabbit polyclonal MEPCE antibody | Novus | NBP2-34858 | WB (1:2000) |
| Antibody | Rabbit polyclonal LARP7 antibody | Abcam | ab-134757 | WB (1:2000) |
| Antibody | Mouse monoclonal HEXIM1 antibody | Santa Cruz Biotechnology | sc-390059 | WB (1 μg/ml) |
| Antibody | Mouse monoclonal actin antibody | Santa Cruz Biotechnology | sc-8432 | WB (1 μg/ml) |
| Antibody | Mouse monoclonal CDK9 antibody | Santa Cruz Biotechnology | sc-13130 | WB (1 μg/ml) |
| Antibody | Mouse monoclonal His-probe antibody | Santa Cruz Biotechnology | sc-53073 | WB (1 μg/ml) |

## Protein expression and purification

The cDNA corresponding to gene of wild-type JMJD6 (1–343), inactive mutant JMJD6 (1–343) H187A/D189A/H273A/K204A/N287A, HEXIM2, and LARP7 was cloned into a pET28a vector containing an N-terminal His$^6$ tag. The DNA corresponding to gene of JMJD6 (1–403) wild-type was cloned into a pGEX vector containing an N-terminal GST tag and TEV linker. All proteins were expressed in Rosetta (DE3) *Escherichia coli* cells. All cell cultures were grown to $A_{600}$ value equal to about 1.0 and induced with a final concentration of 1.0 mM isopropyl 1-thio-β-D-galactopyranoside overnight at 16°C. Cells were resuspended in nickel-binding buffer (50 mM Tris-HCl, pH8.0, 1M NaCl, 1 mM PMSF) and lysed using a sonicator (Fisher Scientific Sonic Dismembrator Model 500) at 35% power, 10 s ON, 5 s OFF, for 20 min. The lysate was centrifuged at 16,000 rpm at 4°C for 30 min. The supernatant corresponding to His-JMJD6 (1–343) was loaded to 7 mL of Ni-NTA resin (GE Healthcare), washed with nickel-binding buffer containing 20 mM imidazole, and eluted with nickel-binding buffer containing 500 mM imidazole. The supernatant corresponding to GST-tev-JMJD6 (1–403) was loaded to 7 mL of glutathione agarose resin (Thermo Scientific), washed with nickel-binding buffer, and eluted with nickel-binding buffer containing 30 mM glutathione. All eluted JMJD6 products were concentrated and purified on a superdex 200 10/300 GL column (GE Healthcare) previously equilibrated with nickel-binding buffer containing 15 mM β-mercaptoethanol. The cDNA fragment encoding full-length wild-type MePCE with an N-terminal His$^6$-tag, full-length mutant MePCE (R169D/R170N/R171D/R172N) with an N-terminal His$^6$-tag, and HEXIM1 with an N-terminal His$^6$-tag was cloned under control of the polyhedrin promoter into a previously described baculovirus transfer plasmid (*Kozono et al., 1994*). Recombinant virus was made by co-transfection into SF9 insect cells (Invitrogen) of the plasmid and BacVector3000 baculovirus DNA (Novagen) using the calcium phosphate co-precipitation method. High titer virus stock was prepared by infection of SF9 insect cells. Wild-type MePCE, mutant MePCE, and HEXIM1 protein was produced by infection of High Five insect cells (Invitrogen) at high multiplicity of infection. Four days later, the cells were lysed by sonication in nickel-binding buffer. Lysates were cleared by centrifugation (20,000 rpm, 60 min) and proteins were purified from the supernatant using 7 mL of Ni-NTA resin (GE Healthcare). The protein was eluted from the column with nickel-binding buffer containing 500 mM imidazole.

## Crystallization, Data Collection, structural determination, and refinement of JMJD6

His-JMJD6 (1–343) was crystallized by vapor diffusion in sitting drops with 0.1M sodium citrate pH 5.6, 1.0M ammonium phosphate monobasic at 8°C. The crystals were soaked in soaking buffer; 0.1M

sodium citrate pH 5.6, 1.0M ammonium phosphate monobasic, 10 mM mono-methyl arginine, 3.75 mM $\alpha$KG, 3.75 mM Iron(II) sulfate. For data collection, His-JMJD6 (1–343) crystals were transferred to a cryo-protecting buffer (soaking buffer supplemented with 25% glycerol (v/v)) and frozen in liquid nitrogen. All data used in structure solving and refinement were collected on a beam line 4.2.2 (MBC-ALS) at the Advanced Light Source (Berkeley, ALS, USA). Data were integrated and scaled using the HKL2000 suite of programs. Structural determination and refinement results are shown in *Supplementary file 1*.

## Western blot analysis

To analyze protein levels, wild-type MEF, *Jmjd6* knockout (KO) MEF, wild-type JMJD6 overexpression in *Jmjd6* KO MEF, inactive mutant JMJD6 overexpression in *Jmjd6* KO MEF, wild-type MePCE overexpression in above four MEF cell lines, mutant MePCE (R169A/R170A/R171A/R172A) overexpression in wild-type JMJD6 overexpression in *Jmjd6* KO MEF, wild-type MePCE with a C-terminal His[6] tag overexpression in *MePCE* KO 293T, and mutant MePCE (R169A/R170A/R171A/R172A) with a C-terminal His[6] tag overexpression in *MePCE* KO 293 T cells were grown on 10 cm plates to be harvested and lysed using a standard RIPA buffer mixed with cOmplete Protease inhibitor cocktail (Roche). MEFs were generated from *Jmjd6*[tm1.1Gbf] knockout mice as previously described (*Böse et al., 2004*; *Hahn et al., 2010*). Total cellular extracts in the presence of a protein standard (Bio-rad) were resolved by 8–12% gradient SDS-PAGE and transferred to a 0.22 µm nitrocellulose membrane and incubated with specific antibodies overnight at 4°C. Antibodies used in this investigation were: Anti-JMJD6 (Santa Cruz Biotechnology, sc-28348), Anti-LARP7 (Abcam, ab-134757), Anti-HEXIM1 (Santa Cruz Biotechnology, sc-390059), Anti-MePCE (Bethyl Laboratories Inc, A304-184A), Anti-Actin (Santa Cruz Biotechnology, sc-8432), Anti-His (Santa Cruz Biotechnology, sc-53073), Anti-Pol II CTD (Gift from Dr. David Bentley), Anti-Pol II Ser2p-CTD (Gift from Dr. David Bentley), and Anti-CDK9 (Santa Cruz Biotechnology sc-13130).

## In vitro MePCE cleavage assay

Full-length MePCE protein with N-terminal His-tag, mixed with EDTA-free cOmplete protease inhibitor (Roche), $\alpha$KG, $Zn^{2+}$, and HEPES pH 6.5, was titrated with recombinant wild-type JMJD6 and placed in 37°C for 2 hr. The reaction was subject to western blot analysis using monoclonal anti-His antibody (Santa Cruz Biotechnology, sc-53073). The reaction was reproduced in two separate experiments and the blot bands were quantified using ImageJ.

## Mass spectrometry

MePCE (161-179) R171-me2s/C177S peptide, mixed with EDTA-free cOmplete protease inhibitor (Roche), $\alpha$KG, $Zn^{2+}$, and HEPES pH 6.5, was treated with recombinant wild-type JMJD6, inactive mutant JMJD6, or peptide alone and placed in 37°C for 2 hr. 1 µL of reaction sample is mixed with 1 µL of a-cyano-4-hydroxycinnamic acid (10 mg/ml in 50% ACN, 0.1% TFA). The mixture is spotted on the MALDI target and allowed to air dry. The sample is analyzed by a Microflex-LRF mass spectrometer (Bruker Daltonics, Billerica, MA) in positive ion reflector mode. External calibration is done using a peptide calibration mixture (4 to 6 peptides) on a spot adjacent to the sample. The raw data is processed in the FlexAnalysis software (version 3.4.7, Bruker Daltonics) and exported in mzXML format. The mzXML files were analyzed on ProteoWizard. Data points were normalized to the intensity of the undigested peptide input and plotted on SigmaPlot v11.0. The reaction was reproduced in three separate experiments.

## Fluorescence polarization experiment

Given the known structure of JMJD6, where aromatic residues are present in close proximity to the binding pocket (Y131, F133, W174, W272), the following tryptophan fluorescence assay was deemed suitable for characterizing various peptide binding activity. All the regular and methylated peptides were synthesized by AnaSpec Inc (Histone 3) or Peptide 2.0 Inc (MePCE). 5 µM JMJD6 (1–343) was titrated and equilibrated with fixed concentrations of each peptide respectively, incubated at 25°C for 30 min between each titration intervals, and subject to fluorescence measurement. The buffer used in the fluorescence quenching assay was 100 mM NaCl, 20 mM Tris-HCl pH 6.5, and 0.05% Tween-20. The excitation wavelength of 280 nm and the emission wavelength of 342 nm was used

for data collection and recorded with a Fluoromax-3 spectrometer. The titration samples were prepared and analyzed in parallel as duplicates, triplicates, or quadruplicates. All values at different titration points were compiled, normalized against the maximum value obtained prior to titration and averaged. The error bars indicate the normalized minimum and maximum values at any given titration point. The $K_D$ for each peptide was calculated by fitting to a four parameter sigmoidal dose-response curve with SigmaPlot v11.0.

### Microscale thermophoresis (MST) experiment

His-JMJD6 (1–343) labeled with fluorescent NT-647 dye at a constant concentration of 100 nM was mixed with sixteen serial dilutions (~1.5 nM–300 µM) of peptides derived from MePCE (154-187), MePCE (154-184) R170-me2s, MePCE (154-184) R171-me2s, Histone 3 (1-21), Histone 3 (1-16) R2me2s, C-peptide (57-87). C-Peptide (57-87) was used as a negative control. MST experiment was performed using Monolith NT.115 (NanoTemper Technologies). His-JMJD6 (1–343) was titrated with peptides in PBS-T buffer (137 mM NaCl, 2.7 mM KCl, 10 mM Na2HPO4, 1.8 mM KH2PO4, and 0.05% Tween-20). The change in the fluorescence of bound and unbound labeled His-JMJD6 (1–343), $\Delta F$, is indicative of the peptide binding. Plotting $\Delta F$ vs. peptide concentration facilitated the generation the dissociation curves, computed by the NTP program. The $K_D$, reflecting the affinity of each of the peptides for His-JMJD6 (1–343), was obtained. The error bars indicate the normalized minimum and maximum values at any given titration point. Each experiment was performed in triplicate or quadruplicate.

### Modeling of JMJD6(1–343)-MePCE(164-178) interaction

Crystal structure of JMJD6 (1–343) monomer (PDB:6MEV) sans methylarginine was used as the template to build in residues corresponding to MePCE (164-178) near the catalytic center of JMJD6 in PyMol. No regard for clashes, bonds, or optimization was considered. Structure was exported in. pdb format and uploaded to YASARA energy minimization server (http://www.yasara.org/minimizationserver.htm) using default parameters (*Krieger et al., 2009*). Energy-minimized output model was converted to. pdb format and the MePCE (164-178) residues were minimally adjusted in PyMol to superimpose R171 of MePCE with the methylarginine observed in the crystal structure (PDB:6MEV).

### Quantification of pol II Ser2p-CTD

Western blot of Pol II Ser2p-CTD and unmodified Pol II CTD in wild-type MEF, *Jmjd6* knockout (KO) MEF, or wild-type JMJD6 overexpression in *Jmjd6* KO MEF was reproduced in four separate experiments. All blot bands were quantified using ImageJ and the band intensity of Pol II Ser2p-CTD was normalized against the band intensity of unmodified Pol II CTD in each respective cell lines. The band intensity of each replicate was normalized against the Pol II Ser2p-CTD level in wild-type MEF, then averaged. The error bars indicate one standard deviation to the observed mean. The P-value was obtained using a paired t-test.

### RNA-Seq

RNAs from wild type MEF cells, MEF cells with *Jmjd6* knockout, and MEF cells with JMJD6 overexpression cells in *Jmjd6* knockout background, respectively, were extracted with Trizol reagent (ThermoFisher Scientific). The extracted RNAs were then sent to Quick biology (Quick Biology, Pasadena) for further mRNA purification using oligo-d(T) beads. The purified mRNA was then used to build a mRNA library. Mouse Genome mm10 was used as the reference.

## Acknowledgements

We thank Dr. David Price from University of Iowa for cDNAs of MePCE and HEXIM1/2, samples of 7SK snRNP complex from Dr. Qiang Zhou at UC Berkeley, Dr. Peter Henson, Dr. James Hagman, Dr. Shaodong Dai, and Dr. Yang Wang, and Janice White for long time support in this project, and other researchers at National Jewish Health (NJH) for their kind support. Binding data was obtained from the Biophysics Core facility at Anschutz Medical Center, University of Colorado at Denver. Mass spectrometry data was obtained from the Proteomics and Metabolomics facility at Colorado State

University. SL is supported by NIH Training grant T32AI007405-28 (to P M). HL is partially supported by NIH Training grant 5T32AI074491-07(to J C). GZ were partially supported by CA201230 (to K B), and NJH bridge fund.

## Additional information

### Competing interests

Gongyi Zhang: has shares in NB Life Laboratory LLC, Colorado. The other authors declare that no competing interests exist.

### Funding

| Funder | Grant reference number | Author |
| --- | --- | --- |
| National Cancer Institute | CA201230 | Kathrin Maria Bernt |
| National Institutes of Health | T32AI007405-28 | Schuyler Lee |
| National Institutes of Health | 5T32AI074491-07 | Haolin Liu |
| National Jewish Health | CA201230 | Gongyi Zhang |

The funders had no role in study design, data collection and interpretation, or the decision to submit the work for publication.

### Author contributions

Schuyler Lee, Data curation, Formal analysis, Methodology, Writing - review and editing; Haolin Liu, Data curation, Formal analysis, Validation, Investigation, Methodology, Writing - review and editing; Ryan Hill, Chunjing Chen, Data curation, Methodology; Xia Hong, Data curation, Investigation; Fran Crawford, Qianqian Zhang, Xinjian Liu, Resources, Data curation, Methodology; Molly Kingsley, Data curation, Writing - review and editing; Zhongzhou Chen, Resources, Data curation, Writing - review and editing; Andreas Lengeling, Data curation, Validation, Writing - review and editing; Kathrin Maria Bernt, Data curation, Funding acquisition, Validation, Writing - review and editing; Philippa Marrack, Funding acquisition, Validation, Visualization; John Kappler, Supervision, Validation; Qiang Zhou, Resources, Data curation, Validation; Chuan-Yuan Li, Yuhua Xue, Resources, Data curation, Validation, Methodology; Kirk Hansen, Data curation, Formal analysis, Validation, Methodology; Gongyi Zhang, Conceptualization, Resources, Data curation, Software, Formal analysis, Supervision, Funding acquisition, Validation, Investigation, Visualization, Methodology, Writing - original draft, Project administration, Writing - review and editing

### Author ORCIDs

Schuyler Lee https://orcid.org/0000-0003-4623-4075
Molly Kingsley http://orcid.org/0000-0002-5921-3743
Kathrin Maria Bernt http://orcid.org/0000-0002-0691-356X
Philippa Marrack http://orcid.org/0000-0003-1883-3687
Qiang Zhou https://orcid.org/0000-0001-7202-3947
Chuan-Yuan Li http://orcid.org/0000-0002-0418-6231
Gongyi Zhang https://orcid.org/0000-0003-3010-3804

### Decision letter and Author response

Decision letter https://doi.org/10.7554/eLife.53930.sa1
Author response https://doi.org/10.7554/eLife.53930.sa2

## Additional files

### Supplementary files

• Supplementary file 1. Crystallographic statistics of complex structure of JMJD6 and a methylated arginine.

• Supplementary file 2. Protein Composition Analysis via Mass Spectrometry. Bacteria expressed and purified JMJD6 are subjected to mass spectrum analysis, all potential contaminated trace protein candidates are listed. No known protease candidate is identified from the list.

• Supplementary file 3. Total RNA-seq reads of wild-type MEF, JMJD6 knockout MEF, and JMJD6 overexpressed in JMJD6 KO background MEF.

• Transparent reporting form

### Data availability

Diffraction data have been deposited in PDB under the accession code 6mev, All data generated or analysed during this study are included in the manuscript and supporting files.

The following dataset was generated:

| Author(s) | Year | Dataset title | Dataset URL | Database and Identifier |
|---|---|---|---|---|
| Lee S, Zhang G | 2019 | Structure of JMJD6 bound to Mono-Methyl Arginine. | https://www.rcsb.org/structure/6mev | RCSB Protein Data Bank, 10.2210/pdb6MEV/pdb |

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
