## [Decision Letter]

**Acceptance summary:**

Your findings on the novel role of JMJD6 on cleaving MePCE and consequently regulating gene transcription are interesting and exciting.

**Decision letter after peer review:**

[Editors’ note: the authors submitted for reconsideration following the decision after peer review. What follows is the decision letter after the first round of review.]

Thank you for choosing to send your work, "JMJD6 Cleaves MePCE to Release P-TEFb", for consideration at *eLife*. Your article has been reviewed by two peer reviewers, including Zhiguo Zhang as the Reviewing Editor and Reviewer #1, and the evaluation has been overseen by a Senior Editor. Although the work is of interest, we regret to inform you that the findings at this stage are too preliminary for further consideration at *eLife*.

Specifically, both reviewers agreed that the study is reporting interesting findings and contained both structural and biochemical data. However, both reviewers thought that additional experiments are needed to support the proposed model, so we decided to reject the manuscript at the current form. However, we would be happy to reconsider the manuscript if you could address major concerns listed from 1 to 4 experimentally as well as other concerns with the text.

*Reviewer #1:*

In this manuscript, Lee et al. studied the role of JMJD6 in the release of P-TEFb, a protein involved in the regulation of RNA polymerase II activity. Several activities including histone lysine/arginine demethylase activity were described for JMJD6. Based on the previous findings from the PI's laboratory, Lee et al. show that JMJD6 has a proteolytic activity, likely targeting MePCE, a core component of 7SK snRNP complex. First, based on the structural studies, Lee et al. found that JMJD6 bound methylated arginine. The binding configuration between methylated arginine and JMJD6 likely rules out the possibility that JMJD6 acts as an arginine demethylase. Instead, JMJD6 functions like JMJD5 and JMJD7 as a protease. Based on the in vitro binding and enzymatic assays as well as in vivo analysis the degradation products of MePCE, they show that MePCE instead of histones is a substrate of JMJD6. Overall, this study is interesting and most of data are convincing. I have following suggestions to improve the manuscript.

1) More data is needed to support the model presented in Figure 3G. for instance, mutations at key residues shown in this figure in MePCE should be made and test how JMJD6 cleavages the mutant form MePCE in vitro and in vivo.

2) To support the model of Figure 5, MePCE mutants that cannot be digested by JMJD6 especially R171 should be used: how do MePCE mutants that cannot be digested by JMJD6 affect Pol2 phosphorylation?

*Reviewer #3:*

This manuscript reports an interesting finding that JMJD6 contains protease activity that is methyl-arginine guided and targets MePCE, a component of the 7SK snRNP, in vitro. The strength of the manuscript is the nice biochemical study of in vitro expressed MePCE and JMJD6 in Figure 2. However, this finding remains largely disjointed from the rest of the manuscript, which left me overall a bit underwhelmed especially with respect to the biological data.

1) The structure of JMJD6 binding to a monomethyl-arginine derivative in Figure 1. Later in Figure 3, dimethyl-arginine in MePCE is critical for binding and cleavage. A structure of a dimethyl-arginine-MePCE peptide would be much more informative.

2) The finding that most inhibitory components of the 7SK snRNP including MePCE itself are positively regulated by JMJD6 via an unknown mechanism is confusing (Figure 2A). This introduces a huge confounding element that remains unexplained but could have direct influence on the phenotype in Figure 4.

3) The model in Figure 4 does not mention MePCE cleavage and is exclusively composed of elements from other publications featuring the recruitment of JMJD6 via RNA and binding to BRD4. No experimental data in the manuscript supports this model. In fact, this model can function without MePCE cleavage.

4) The recognition and cleavage site mapping in MePCE is nice but would be even nicer if supported by mutagenesis of full-length MePCE and testing in vitro and after overexpression in cells.

5) The S2P phenotype in Figure 4 is not the cleanest and should be quantified in multiple experiments. No evidence links this phenotype to the described cleavage of MePCE. In fact, a recent paper by Shelton et al., 2018 questions that MePCE is indeed part of the inhibitory 7SK snRNP but activates P-TEFb in a BRD4-independent manner. This paper should be discussed and the finding should be linked to MePCE cleavage, i.e. by looking at 7SK snRNP large complex levels in the JMJD6 KO cells.

[Editors’ note: further revisions were suggested prior to acceptance, as described below.]

Thank you for resubmitting your work entitled "JMJD6 Cleaves MePCE to Release P-TEFb" for further consideration by *eLife*. Your revised article has been reviewed by two peer reviewers, including Zhiguo Zhang as the Reviewing Editor and Reviewer #1, and the evaluation has been overseen by Jessica Tyler as the Senior Editor.

The manuscript has been improved but there are some remaining issues that need to be addressed before acceptance, as outlined below:

1) Please edit some minor writing issues.

2) Please provide statistical analysis for Figure 4B and include MeCPE in their model in Figure 4D (as you are missing the point of the paper by not highlighting the main finding -> JMJD6 cleaves MeCPE and activates P-TEFb in their model.)

---

## [Author Response]

[Editors’ note: the authors resubmitted a revised version of the paper for consideration. What follows is the authors’ response to the first round of review.]

Reviewer #1:[…] Overall, this study is interesting and most of data are convincing. I have following suggestions to improve the manuscript.

*1) More data is needed to support the model presented in Figure 3G. for instance, mutations at key residues shown in this figure in MePCE should be made and test how JMJD6 cleavages the mutant form MePCE* in vitro *and* in vivo.

Please see our response to reviewer #3, comment #3.

2) To support the model of Figure 5, MePCE mutants that cannot be digested by JMJD6 especially R171 should be used: how do MePCE mutants that cannot be digested by JMJD6 affect Pol2 phosphorylation?

We have added data in Figure 3 and Figure 4 to address this question (see our response to reviewer #3, comment #3.

Reviewer #3:
*This manuscript reports an interesting finding that JMJD6 contains protease activity that is methyl-arginine guided and targets MePCE, a component of the 7SK snRNP,* in vitro*. The strength of the manuscript is the nice biochemical study of* in vitro expressed MePCE and JMJD6 in Figure 2. However, this finding remains largely disjointed from the rest of the manuscript, which left me overall a bit underwhelmed especially with respect to the biological data.1) The structure of JMJD6 binding to a monomethyl-arginine derivative in Figure 1. Later in Figure 3, dimethyl-arginine in MePCE is critical for binding and cleavage. A structure of a dimethyl-arginine-MePCE peptide would be much more informative.

Crystals of JMJD6 took more than one year to grow, we even had a hard time to repeat this crystallization condition in the past year, it will be our future focus to get complex structures of JMJD6 with different substrates.

2) The finding that most inhibitory components of the 7SK snRNP including MePCE itself are positively regulated by JMJD6 via an unknown mechanism is confusing (Figure 2A). This introduces a huge confounding element that remains unexplained but could have direct influence on the phenotype in Figure 4.

As we mentioned in the context, it is a big puzzle for us to interpret the result so far since we did not see any change of mRNA in the JMJD6 knockout cell line. It must be some translational issue. It is one of our next question to address. For Figure 4, we have added more convincing data now, see below.

3) The model in Figure 4 does not mention Mope cleavage and is exclusively composed of elements from other publications featuring the recruitment of JMJD6 via RNA and binding to BRD4. No experimental data in the manuscript supports this model. In fact, this model can function without MePCE cleavage.

We have added new data to support the Figure 4 model, in which we show the relative position of JMJD6 and how it is recruited by RNA and BRD4. As you can see from Figure 4A, knockout of JMJD6 lead to drastic loss of active Pol II (Pol II Ser2p-CTD) whereas JMJD6 overexpression leads to a drastic increase in active Pol II. The newly added Figure 4B showed that Pol II Ser2p-CTD dropped ~50% in three repeats when JMJD6 is knocked out, whereas Pol II Ser2p-CTD is increased to ~150% when JMJD6 is overexpressed. Newly added Figure 4C showed that MePCE knockout dramatically increases the content of active Pol II (Pol II Ser2p-CTD). These new data confirm that both JMJD6 and MePCE are closely related to the activities of Pol II.

We have added mutation data to support the model of Figure 3G now. Our new collaborator (author added), Dr. Yuhua Xue’s group has generated MePCE knockout 293T cell line as demonstrated in Figure 3H. When we introduced wild type of MePCE into this MePCE KO cell line (Figure 3I), we saw a degraded band (Figure 3I, left lane), but not the mutant MePCE with –RRRR- to –AAAA- change (Figure 3I, right lane). On the other hand, it remains true that when the two versions are introduced to the JMJD6 overexpression cell line with JMJD6 knockout background (Figure 3J), wild type MePCE has two bands (Figure 3J, left), while mutant MePCE has a dominated band but a weak low molecular band. We believe that the high content of overexpressed JMJD6 could lead to weak non-specific cut at the mutant site. in vitro data is shown in Figure 3—figure supplement 2B (native MePCE), and Figure 3—figure supplement 2C (mutant MePCE).

*4) The recognition and cleavage site mapping in MePCE is nice but would be even nicer if supported by mutagenesis of full-length MePCE and testing* in vitro *and after overexpression in cells.*

Please see our response to comment #3 above.

5) The S2P phenotype in Figure 4 is not the cleanest and should be quantified in multiple experiments. No evidence links this phenotype to the described cleavage of MePCE. In fact, a recent paper by Shelton et al., 2018 questions that MePCE is indeed part of the inhibitory 7SK snRNP but activates P-TEFb in a BRD4-independent manner. This paper should be discussed and the finding should be linked to MePCE cleavage, i.e. by looking at 7SK snRNP large complex levels in the JMJD6 KO cells.

We have added new data in Figure 4, three repeats showed the drastic dropping of active Pol II (~50%,Pol II Ser2p-CTD) when JMJD6 is knocked out, whereas a drastic increase of active Pol II (~150%,Pol II Ser2p-CTD) is observed when JMJD6 is overexpressed (Figure 4B). At the same time, knockout of MePCE just works opposite and dramatically increase the content of active Pol II in an 293T MePCE knockout cell line (Figure 4C). All these new data support our model of the role of JMJD6 in helping BRD4 to recruit CDK9 to Pol II (Figure 4D).

There is an interesting finding showing an alternative or free form of MePCE that could recruit CDK9 directly in Shelton et al.’s 2018 report. This form may work on a small number of gene targets as the authors mentioned, but overall MePCE should play an inhibitory role on CDK9. A recent nonsense mutant in a human disease, which increase the activity of CDK9 or Pol II, also support this general role of MePCE (Schneeberger et al., 2019). Both are cited in our revision now. Of interest, Shelton et al., found that CDK9 interacts with the N-terminal domain of MePCE (residues 1-400), it is possible that MePCE acts as an anchor to help 7SK snRNP to fast CDK9. We found that there is a similar cutting site around residues 363-366 (-RKRRR-) compared to our identified site around residues 167-172 (-KRRRR-) though we did not detect fragments cut from this site from our in vitro experiments so far, this may be an interest topic needing further investigation. We discuss it in the revised version now. The N-terminal portion of MePCE is glycine rich, which makes it a big challenge for expression and characterization.

The effect of JMJD6 on stability of 7SK snRNP was very well characterized by Drs. Michael Rosefeld and Wei Liu’s groups (Liu et al., 2013). They found that JMJD6 disrupts the 7SK snRNP complex, we have discussed it in the context.

[Editors’ note: what follows is the authors’ response to the second round of review.]

Thank you for resubmitting your work entitled "JMJD6 Cleaves MePCE to Release P-TEFb" for further consideration by eLife. Your revised article has been reviewed by two peer reviewers, including Zhiguo Zhang as the Reviewing Editor and Reviewer #1, and the evaluation has been overseen by Jessica Tyler as the Senior Editor.The manuscript has been improved but there are some remaining issues that need to be addressed before acceptance, as outlined below:1) Please edit some minor writing issues.

We have edited the manuscript very carefully to reduce the spelling, grammar, and others.

2) Please provide statistical analysis for Figure 4B and include MeCPE in their model in Figure 4D (as you are missing the point of the paper by not highlighting the main finding -> JMJD6 cleaves MeCPE and activates P-TEFb in their model.)

Statistical analysis for Figure 4B has been provided and is described in details in the “Materials and methods”/“Quantification of Pol II Ser2p-CTD” section and figure legend. Figure 4D is newly added to highlight the model of JMJD6 cleaving MePCE in the context of 7SK snRNP disruption and P-TEFb release, as well as provide further insight via a surface charge model of JMJD6 and it’s known interactions with BRD4 and ssRNA, in accordance with reports from past publications.